# Production, Passaging Stability, and Histological Analysis of Madin–Darby Canine Kidney Cells Cultured in a Low-Serum Medium

**DOI:** 10.3390/vaccines12090991

**Published:** 2024-08-30

**Authors:** Ming Cai, Yang Le, Zheng Gong, Tianbao Dong, Bo Liu, Minne Su, Xuedan Li, Feixia Peng, Qingda Li, Xuanxuan Nian, Hao Yu, Zheng Wu, Zhegang Zhang, Jiayou Zhang

**Affiliations:** 1Wuhan Institute of Biological Products Co., Ltd., Wuhan 430207, China; 17691160247@163.com (M.C.); 13135687868@163.com (Y.L.); gz937161787@163.com (Z.G.); liubohust@126.com (B.L.); xuedanli@whu.edu.cn (X.L.); pfx12706695702023@163.com (F.P.); 18108676458@163.com (Q.L.); nianxuanxuan@126.com (X.N.); 18062428392@163.com (H.Y.); zheng_9837@163.com (Z.W.); 2National Engineering Technology Research Center for Combined Vaccines, Wuhan 430207, China; 3National Key Laboratory for Novel Vaccines Research and Development of Emerging Infectious Diseases, Wuhan 430207, China; 4Hubei Provincial Vaccine Technology Innovation Center, Wuhan 430207, China; 5Center for Drug Evaluation and Inspection of HMPA (Hubei Center for Vaccine Inspection), Wuhan 430207, China; dongtianbao@foxmail.com

**Keywords:** MDCK cells, low-serum medium, RNA sequencing, proteomics, flu vaccine

## Abstract

Madin–Darby canine kidney (MDCK) cells are commonly used to produce cell-based influenza vaccines. However, the role of the low-serum medium on the proliferation of MDCK cells and the propagation of the influenza virus has not been well studied. In the present study, we used 5 of 15 culture methods with different concentrations of a mixed medium and neonatal bovine serum (NBS) to determine the best culture medium. We found that a VP:M199 ratio of 1:2 (3% NBS) was suitable for culturing MDCK cells. Furthermore, the stable growth of MDCK cells and the production of the influenza virus were evaluated over long-term passaging. We found no significant difference in terms of cell growth and virus production between high and low passages of MDCK cells under low-serum culture conditions, regardless of influenza virus infection. Lastly, we performed a comparison of the transcriptomics and proteomics of MDCK cells cultured in VP:M199 = 1:2 (3% NBS) with those cultured in VP:M199 = 1:2 (5% NBS) before and after influenza virus infection. The transcriptome analysis showed that differentially expressed genes were predominantly enriched in the metabolic pathway and MAPK signaling pathway, indicating an activated state. This suggests that decreasing the concentration of serum in the medium from 5% to 3% may increase the metabolic activity of cells. Proteomics analysis showed that only a small number of differentially expressed proteins could not be enriched for analysis, indicating minimal difference in the protein levels of MDCK cells when the serum concentration in the medium was decreased from 5% to 3%. Altogether, our findings suggest that the screening and application of a low-serum medium provide a background for the development and optimization of cell-based influenza vaccines.

## 1. Introduction

Influenza viruses are highly prone to causing pandemics owing to their rapid mutation, infectiousness, and pathogenicity, thus posing a significant threat to public health and human lives worldwide. Annual flu vaccination is an effective measure to prevent potential flu pandemics [1]. Presently, influenza vaccine production in China heavily relies on the use of chicken embryos [2,3]. Nevertheless, this method has several limitations, such as a lengthy production cycle, complex operations, and potential issues with egg supply adequacy [4]. Cell-based influenza vaccines are expected to become the popular method for the development of future influenza vaccines. Madin–Darby canine kidney (MDCK) cells have been used as a continuous cell line for influenza vaccine production [5,6]. They are widely used for the cultivation of different subtypes of influenza viruses [7].

Neonatal bovine serum (NBS) is added to the culture medium during the cell culture phase of cell-based influenza vaccine production, at a concentration of 5–10% (*V*/*V*). NBS addition increases cell attachment and growth [8]. On the other hand, serum amyloid-like protein P inhibits salivary acylated glycoproteins on the surface of influenza A viruses, thus affecting viral production. Excess NBS can cause exogenous contamination, which can be costly and unsuitable for downstream product purification, among other limitations. This limitation can be addressed by adding a mixed serum to decrease the concentration of serum, or even using a serum-free medium, which is also a future direction for cell-based vaccine production [9].

The effect of a low-serum medium and mixed-serum medium (VP-SFM/M199) on MDCK cell proliferation, viral infectivity, and manufacturing processes has not been thoroughly investigated. Therefore, in the present study, we investigated the proliferation ability and influenza virus infectivity of MDCK cells in a low-serum culture medium using the cell culture method described in our previous study. Changes in different treatment groups will be analyzed using genomic data, comparing trends at the transcriptional, protein, and biological pathway levels. The goal is to establish a foundation for the application of a low-serum culture medium in cell-based influenza vaccine production.

## 2. Materials and Methods

### 2.1. Cells and Virus

MDCK cells, obtained from the ATCC stock (ATCC CCL-34), were cultured using the adherent culture method in M199 supplemented with NBS. The main cell bank (MDCK-M-60, P60) and the working cell bank (MDCK-W-63, P63) were established by the Department of Viral Vaccine Research II at the Wuhan Institute of Biological Products, following the requirements of the Chinese Pharmacopoeia, Third Part (2020 Edition). Influenza viruses, namely A/Victoria/4897/2022 (H1N1), A/Darwin/9/2021 (H3N2), BVR-26 (B-Victoria Lineage), and BVR-1B (B/Phuket/3073/2013), were maintained in the Viral Vaccine Research Unit II of the institute.

### 2.2. Study of MDCK Cell Culture and Production Process

#### 2.2.1. Screening for Low-Serum Media

MDCK cells grown into dense monolayers in T75 flasks were digested with 0.25% trypsin-EDTA (Gibco, Grand Island, NE, USA). The digestion was inhibited with the addition of media, and the number of cells was counted. The cells were then inoculated into T75 flasks at a cell density of 8 × 10^4^ cells/cm^2^ (cell density of T75 flasks = 600 × 10^4^ cells/flask). The cells were cultured in an incubator at 37 °C with 5% CO_2_, using different ratios of culture medium (VP-SFM/M199 = 1:2, 1:3, and 1:4) with varying concentrations of NBS ranging from 1% to 5%. The cells were incubated in a CO_2_ incubator, and the cell morphology was evaluated daily at the same time. The cells were harvested and counted after 24 and 144 h, and cell growth curves were plotted.

#### 2.2.2. Validation of Scale-Up Culture

The cells were cultured to form a monolayer using the culture method using the best culture effect from experiment 1.1. The cells were then passaged onto the cell factories (CF1, CF2, and CF10) and incubated in a 5% CO_2_, 37 °C incubator for amplifying the culture for 72 h. The growth of the cells in each group was observed, and the cells were digested as they grew into a dense monolayer. Sampling and counting were performed to determine the optimal culture method. The control group contained a mixed medium (VP-SFM:M199 = 1:2) containing 5% NBS.

#### 2.2.3. The Cell Production Process in a Low-Serum Medium

The cells were initially revived in T75 flasks. Once they reached an optimal growth status and formed a monolayer, they were transferred to cell factories (CF2, CF10) at a 1:15 passage ratio. The cell factories were then placed in an incubator set at 37 °C with 5% CO_2_. After 72 h, trypsin was used for wet-washing and digestion, which was stopped by adding a medium containing an appropriate proportion of NBS. The inoculum density of 80 × 10^4^ cells/mL was then transferred to a 7.5 L bioreactor for further incubation. Tablet carriers (Fibra-Cel^®^, Eppendorf, Hamburg, Germany) and phosphate-buffered saline solution were added to the bioreactor and sterilized using moist heat before use. The culture parameters were as follows: pH 7.2, 60% dissolved oxygen, a stirring paddle speed of 60 rpm, and a temperature of 37 °C. After cell inoculation, the fresh medium was replenished via perfusion and the glucose concentration in the medium was maintained within the range of 8.0–15.0 mmol/L by adjusting the perfusion rate. The glucose concentration of the culture medium was frequently monitored. The cell suspension was then removed and counted using 0.2% Trypan blue staining. The control group contained a mixed medium (VP-SFM:M199 = 1:2) with 5% NBS.

### 2.3. Passaging Stability of MDCK Cells in a Low Serum Medium

#### 2.3.1. Stability of Continuously Passaged MDCK Cell Culture

MDCK cells were revived and cultured in T75 flasks at 37 °C with 5% CO_2_. Cells showing robust growth and forming a confluent monolayer were subsequently passaged (P64–P80) in a low-serum medium at a 1:15 ratio. The culture supernatant and cell suspension were stained with 0.2% Trypan blue, and the cell density, viability, and metabolic levels of cells cultured in a low serum medium were continuously evaluated. The control group contained a mixed medium (VP-SFM:M199 = 1:2) containing 5% NBS.

#### 2.3.2. Characterization of MDCK Cell Development across Generations

MDCK cells cultured in low-serum medium from passages P65, P70, P75, and P80 were inoculated at three initial densities: 5 × 10^4^ cells/mL, 10 × 10^4^ cells/mL, and 15 × 10^4^ cells/mL and incubated in a 37 °C, 5% CO_2_ incubator. Furthermore, the cell number and viability were determined and plotted for analysis. Cell counting and viability assessment were performed using 0.2% Trypan blue staining at 0 to 144 h, and growth curves were generated. To evaluate the clonogenic potential of the individual cells, MDCK cells in the logarithmic growth phase from different passages were used in a plate cloning assay [10]. The number of visible clones was quantified using Image J software(v1.8.0.345).

#### 2.3.3. Virus Susceptibility Test

A/Victoria/4897/2022 (H1N1), A/Darwin/9/2021 (H3N2), BVR-26 (B-Victoria Lineage), and BVR-1B (B/Phuket/3073/2013) were evaluated on MDCK cell line at passages 65, 70, 75, and 80. In the influenza virus susceptibility tests, three parallel sets of samples were incubated at 34 °C with 5% CO_2_ for durations ranging from 48 to 120 h to monitor viral replication. Viral supernatants from each group were collected to assess hemagglutination efficacy. The control group was grown in a mixed medium (VP-SFM:M199 = 1:2) with 5% NBS, and no viral maintenance solution was added.

### 2.4. Transcriptomic and Proteomic Sequencing

#### 2.4.1. RNA Extraction and RNA-Seq

The TaKaRa MiniBEST Universal RNA Extraction Kit (Takara, Shiga, Japan) was used to extract the total cellular RNA, which was then sent to Wuhan Biobank (Wuhan, China) for RNA sequencing. The process involved cDNA synthesis, library preparation, amplification, quality control, sequencing, and subsequent data analysis. Differentially expressed genes (DEGs) were identified using criteria of |log2(FoldChange)| ≥ 1 and Padjust ≤ 0.05. A Kyoto Encyclopedia of Genes and Genomes (KEGG) pathway enrichment analysis was performed for DEGs (Appendix A).

#### 2.4.2. Proteomic Analysis

The harvested cells were submitted to Wuhan Biobank (China) for proteomic analysis. The workflow involved sample processing, mass spectrometry detection, database retrieval, data quality control, data standardization, differential protein screening, and enrichment analysis. Detailed protocols and procedures are mentioned in Appendix A.

## 3. Results

### 3.1. Study on the Culture Process of MDCK Cells in Low Serum Medium

After evaluating various methods to cultivate MDCK cells, we found that cell density increased with higher NBS and VP ratio concentrations. The cell growth rate was slower when using a mixed medium with 1–2% serum concentration, and the logarithmic phase was delayed. When a mixed medium containing 3% serum concentration was used, the cells exhibited robust growth with clear boundaries and formed a dense monolayer without cell detachment or death. As shown in Figure 1, the growth status of MDCK cells in five cultures (VP:M199 = 1:2 (2% NBS), VP:M199 = 1:3 (3% NBS), VP:M199 = 1:4 (3% NBS), VP:M199 = 1:2 (3% NBS), and VP:M199 = 1:3 (4% NBS)) were selected for validation of the cell culture expansion in CF1, CF2, and CF10.

As shown in Figure 2, MDCK cells cultured in the CF1 and CF2 groups exhibited robust growth and tight alignment, with stable cell growth during passaging. At a specific concentration of NBS, the density of MDCK cells after expansion culture increased with a higher ratio of VP-SFM medium. The VP:M199 = 1:2 mixed medium exhibited the most favorable effect. When the medium ratio was fixed, no significant difference was found in terms of cell density among cells cultured in different sizes of cell factories with 3% and 5% serum concentrations. Nevertheless, the number of cells was significantly higher in the 5% serum medium than in the 2% serum medium. This observation suggests that when NBS is decreased to 3%, the culture effect on the MDCK cells is equal to or more than that at 5% (NBS). Therefore, VP:M199 = 1:2 (3% NBS) can be considered the most suitable method for the in vitro culture of MDCK cells. Hence, VP:M199 = 1:2 (3% NBS) was selected for subsequent experiments using the low-serum culture method.

### 3.2. Cell Culture in a Low-Serum Medium Bioreactor

CF10 was digested to obtain seed pots for inoculation in the bioreactor, with an inoculum density of 80 × 10^4^ cells/mL (Table 1). After 72 h of incubation, the cell density of the low-serum medium group reached 410 × 10^4^ cells/mL at a VP:M199 ratio of 1:2 (3% NBS), whereas the control group showed a cell density of 408 × 10^4^ cells/mL. No significant difference was observed between the two culture types. The control group showed a density of 408 × 10^4^ cells/mL, and no significant difference was observed between the two culture methods. Cells cultivated in the low-serum medium were less differentiated and showed a better cellular state with fewer dead cells. Therefore, low serum levels could meet the requirements of both MDCK cell culture and amplification culture, and simultaneously could reduce the influx of cell contaminants caused by serum components.

### 3.3. Stability Study of MDCK Cell Passaging

MDCK cells (P64) were revived in a low-serum medium. Both low-serum and control media effectively promoted MDCK cell growth and proliferation even after 15 generations of consecutive passaging cultures. Their densities and viability ranged from 110 × 10^4^ cells/mL to 150 × 10^4^ cells/mL and from 80% to 90%, respectively, and cells in both groups showed flat and regular cell morphology (Figure 3A,B). Consistent trends were observed for the rates of glucose-specific consumption and lactate-specific synthesis of cell culture supernatants in the two groups.

The P65, P70, P75, and P80 of MDCK cells were injected with varying cell densities in low-serum media to encourage normal MDCK cell growth, and the growth curves were all “S” shaped (Figure 4A–D). The cell growth pattern was generally consistent as cell generation increased, and the higher the inoculation density, the more cells were in the initiation phase, with a logarithmic growth period ranging from 48 h to 96 h before entering the plateau phase. Moreover, both cell groups developed an appropriate number of clonal colonies (Figure 4E,F). The average number of colonies established in P65 was 155 and 150, in P70 was 172 and 165, in P75 was 163 and 168, and in P80 was 154 and 154, with no notable variation in colony numbers. Finally, virus sensitivity experiments revealed that MDCK cells cultured in P65, P70, P75, and P80 in low-serum culture mode and a control were slightly less sensitive to influenza A viruses (H1N1, H3N2) than to influenza B viruses (BV, BY), and that virus sensitivity decreased as the number of cell generations increased. The results showed that, when considering the cost of cell culture and the pressure of downstream purification, low serum concentrations were equally suitable for influenza virus proliferation on MDCK cells (Figure 5A–D). Furthermore, there was no discernible effect of culturing cells in low serum medium on the ability of influenza virus to infect MDCK cells.

### 3.4. Transcriptomic and Proteomic Analysis of MDCK Cells Cultured in Medium with 3% and 5% Serum

To further investigate the impact of serum concentration changes on MDCK cells, MDCK cells were cultured in a separate cell culture medium containing 3% and 5% serum (designated as 3-uninfected and 5-uninfected, respectively). Subsequently, these two cell groups were infected with H1N1 influenza virus (designated as 3-infected and 5-infected). Finally, transcriptomic and proteomic analyses were performed on these four cell groups.

The transcriptomics analysis revealed that out of the 31,121 examined genes, a total of 654 differentially expressed genes (DEGs) were identified in the comparison between the 3-uninfected group and the 5-uninfected group, accounting for 2.1% of the detected genes. Among these DEGs, 272 genes exhibited upregulation while 382 genes exhibited downregulation (Figure 6A). In contrast, a total of 520 DEGs were observed between the infected groups (3-infected vs. 5-infected), accounting for 1.7% of the detected genes. Among them, 368 genes were upregulated and 152 genes were downregulated (Figure 6B). The proportion of DEGs is generally limited in magnitude. Subsequently, KEGG enrichment analysis was performed on the DEGs, revealing that these DEGs were significantly enriched in a limited number of signaling pathways. The DEGs in the 3-uninfected group, compared to the 5-uninfected group, were found to be significantly enriched in seven signaling pathways according to KEGG enrichment analysis. The most significantly enriched pathways were the metabolic pathway and the MAPK signaling pathway, both of which were activated (Figure 6C). Similarly, the DEGs of the 3-infected group and the 5-infected group exhibited similar results following KEGG enrichment analysis, demonstrating enrichment in six signaling pathways. Notably, the metabolic pathway and the MAPK signaling pathway were also the most significantly enriched activated pathways (Figure 6D).

The proteomic analysis also revealed a limited number of differentially expressed proteins, with only five proteins showing differential expression between the 3-uninfected group and the 5-uninfected group, namely PCM1, HPCAL1, CA2, IL1RAP, and SMYD4 (Figure 6E). Similarly, within the 3-infected group versus the 5-infected group, only four proteins exhibited differential expression: TGM2, TMSB10, CLU, and IFGGB2 (Figure 6F). By performing correlation analysis between transcriptome data and proteome data, we identified only one co-expressed gene (CA2) in the 3-uninfected group compared to the 5-uninfected group (Figure 6G). Similarly, there were only two commonly expressed genes (TGM2 and IFGGB2) in the 3-infected group compared to the 5-infected group (Figure 6H). However, due to the limited number of differentially expressed proteins, further enrichment analysis cannot be conducted. These findings suggest that reducing serum concentration from 5% to 3% in the culture medium has minimal impact on MDCK cells. The relevant data of transcriptome and proteome are provided as Appendix A.

## 4. Discussion

MDCK cells are frequently used for influenza vaccine production as they help maintain the genetic characteristics of the virus and produce high-quality vaccines. They can be cultivated in a reactor for large-scale production to meet vaccine demands during an influenza pandemic [11,12,13,14,15]. Decreased serum concentrations offer many advantages, such as significantly reduced production costs, simplified purification, minimum batch-to-batch variations, and a lowered risk of contamination [16]. Herein, we investigated the effects of the low-serum culture method on MDCK cells to improve the production of cell-matrix influenza vaccines. First, we compared the growth curves and viability of MDCK cells cultured using 15 methods and selected five methods for scale-up. MDCK cell density increased with increasing NBS concentration and VP ratio. The results revealed an NBS concentration of 3% as a threshold. Additionally, it has been demonstrated that VP-SFM medium significantly impacts both small-scale and large-scale cell cultures, as well as virus proliferation [17,18,19,20]. When the concentration decreased below 3%, cell growth slowed down and the logarithmic phase of cell growth was delayed. When no neonatal bovine serum was added, only a few cells adhered to the wall. However, after 48 h of culture, all the cells were completely shed. Conversely, at an NBS concentration of 3%, the cells exhibited robust growth and formed a dense monolayer, thereby creating conditions conducive to influenza virus infection. However, when the concentration exceeded 3%, the cells exhibited accelerated growth. Furthermore, some cells exhibited a blurred and contoured cell morphology owing to their high density and a small amount of cell shedding, thereby creating conditions not conducive to influenza virus infection. Subsequently, under a certain serum concentration, as the proportion of VP-SFM increased in the medium, the density of MDCK cells increased after cell factory expansion culture. Specifically, when MDCK cells were cultured in a medium with 3% and 5% serum concentrations at a VP:M199 ratio of 1:1, no significant difference was observed in cell factories with different specifications. In a 7.5-L bioreactor culture of MDCK cells, VP:M199 ratios of 1:2 (3% NBS) and 1:2 (5% NBS) achieved cell culture densities above 400 × 10^4^ cells/mL. No significant difference was observed, cell culture effect was slightly higher than similar studies.

The stability study conducted using MDCK cells cultured by two different methods, given the critical importance of cell stability in vaccine production, showed that both culture methods yielded MDCK cells with consistent growth, metabolism, and virus sensitivity without any significant difference [21,22,23,24].

Transcriptomics and proteomics technologies have been extensively employed in the realm of vaccine research [25,26,27,28]. The transcriptomics and proteomics results can offer a more comprehensive insight into the impact of reducing the newborn calf serum (NBS) concentration from 5% to 3% on MDCK cell cultures. Previous studies have shown that the appropriate reduction of serum concentration does not significantly inhibit cell growth [29,30,31]. Even at low concentrations, the presence of serum provides excellent cell viability [32]. This study obtained similar results through transcriptomic and proteomic data analysis. There are only a few differentially expressed genes and proteins observed. In particular, the number of differentially expressed proteins does not exceed 10, and further KEGG enrichment analysis did not identify any enrichment of these differentially expressed proteins in any signaling pathway. The results suggest that reducing the NBS concentration in the cell culture medium from 5% to 3% has a negligible influence on MDCK cells. Meanwhile, the KEGG enrichment analysis revealed that these DEGs were exclusively enriched in a limited number of signaling pathways, with the metabolic and MAPK signaling pathways being the most significantly enriched ones, both exhibiting activation. This implies that the reduction in serum concentration may potentially augment the transcriptional expression of genes related to metabolism in MDCK cells. This phenomenon of enhanced metabolism caused by low serum can also be observed in the research conducted by Xiong et al., which indicates that some biosynthesis pathways were upregulated in osteoclasts cultured under low serum conditions [33]. This may be a compensatory metabolic enhancement to meet the demands of cell growth. However, in the present study, the activation of these metabolic signaling pathways was not detected in the proteomic results. The incomplete consistency between transcriptome and proteome results has been commonly observed in previous research findings [34,35]. It may be caused by differences in protein degradation rates, post-transcriptional regulation, and post-translation regulation, which also reflect the complexity of biological processes. We only identified three commonly expressed proteins: CA2, TGM2, and IFGGB2 by performing correlation analysis between transcriptome and proteome data. CA2 can catalyze the reversible hydration of carbon dioxide [36]. TGM2 is an enzyme that catalyzes protein crosslinking by epsilon–gamma glutamyl lysine isopeptide bonds [37]. IFGGB2 is an interferon-inducible GTPase 1-like protein that is associated with viral replication [38]. However, the expression of other proteins in the signaling pathways associated with these three proteins did not show any differences. Therefore, we believe that the impact of individual protein changes on MDCK cells is limited. In addition, the antiviral ability, adhesiveness, and tumorigenicity of MDCK cells are also crucial factors to consider [39,40,41]. KEGG enrichment analysis of differentially expressed genes and proteins showed no significant enrichment in signal pathways associated with antiviral responses, cell adhesion, or tumorigenesis (Appendix A). However, further experimentation is required to validate these findings and whether decreased serum concentrations potentially affect other characteristics of MDCK cells should also be clarified.

In summary, the present study indicates the feasibility of culturing MDCK cells in a low-serum medium for the production of cell-based influenza vaccines. The present findings provide a foundation for improving the production of cell-based influenza vaccines.

## 5. Conclusions

In the production of cell-based influenza vaccines, the use of a low-serum medium can enhance the quality and purity of the product. It also facilitates the separation and purification processes, reduces contamination caused by serum and impurities, and lowers production costs. MDCK cells and virus amplification in bioreactors can be easily carried out, allowing for online monitoring and accurate feeding. In this study, we provide a detailed description of the optimization of the MDCK cell culture method and technology. Specifically, we focus on the low-serum culture medium condition, the stability of extended MDCK cells, and virus sensitivity. We conducted related research to compare MDCK cells cultured under low serum conditions with those in the early stages of the process. Our findings indicate that cell growth is stable and virus sensitivity is strong under low-serum culture conditions. This may be attributed to the easier removal of residual serum from MDCK cells cultured in low serum, which enhances TPCK trypsin’s ability to cleave hemagglutinin protein onto the influenza virus. Furthermore, transcriptome and proteomics analysis revealed that the impact of reducing serum concentration from 5% to 3% on MDCK cells is minimal. However, this reduction in serum concentration could potentially enhance the activation of cell metabolism-related genes at the transcriptional level. In conclusion, this study demonstrates that a low-serum medium can be utilized for the amplification of MDCK cells and the proliferation of the influenza virus. This approach has the potential to reduce costs and increase efficacy, while ensuring high vaccine yield and quality.

## Figures and Tables

**Figure 1 vaccines-12-00991-f001:**
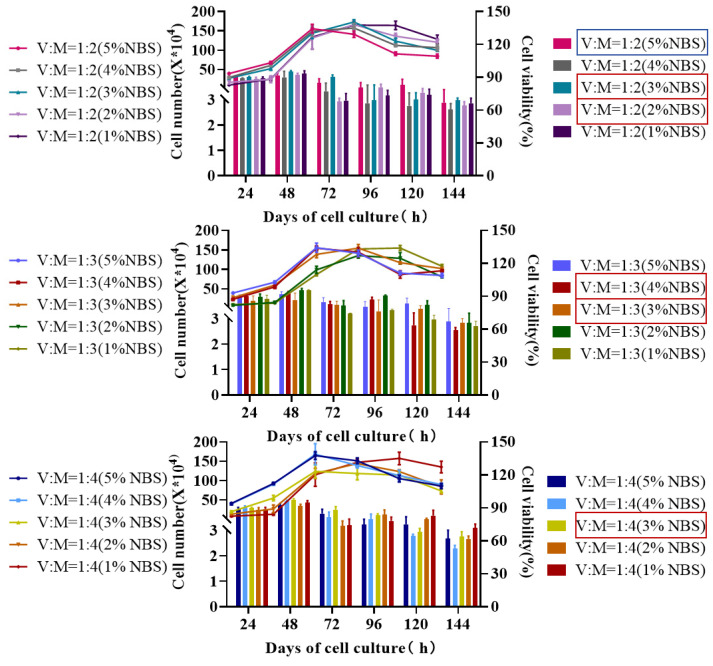
Growth curves of MDCK cells in 15 cultures; VP:M199 = 1:2 (3%NBS, 2% NBS), VP:M199 = 1:3 (4% NBS, 3% NBS), and VP:M199 = 1:4 (3% NBS) cultures were selected for the validation of cell factory scale-up cultures (blue: control, red: experimental).

**Figure 2 vaccines-12-00991-f002:**
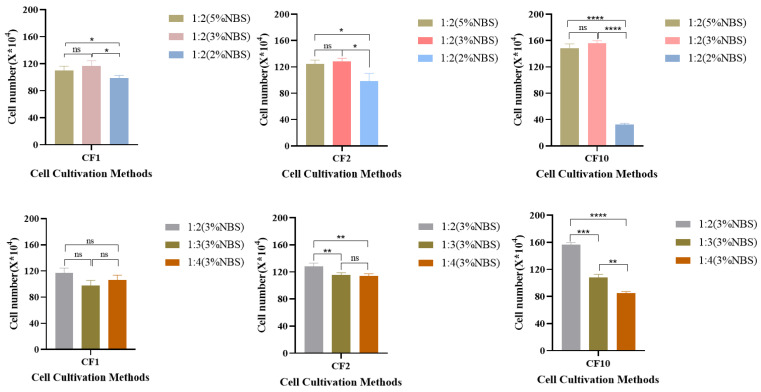
MDCK cell factories (CF1, CF2, CF10) enlarged culture validation results. ns Not Significant, * *p* < 0.05, ** *p* < 0.01, *** *p* < 0.001, **** *p* < 0.0001.

**Figure 3 vaccines-12-00991-f003:**
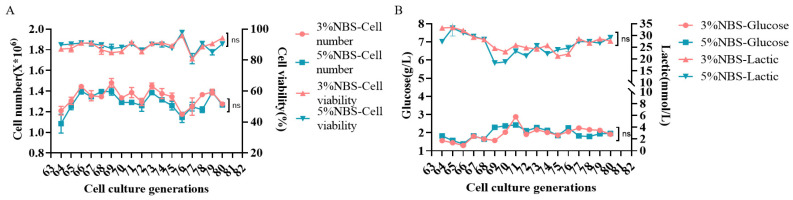
MDCK cell culture continuous passaging stability ((**A**) cell growth status, (**B**) cell metabolism). ns Not Significant.

**Figure 4 vaccines-12-00991-f004:**
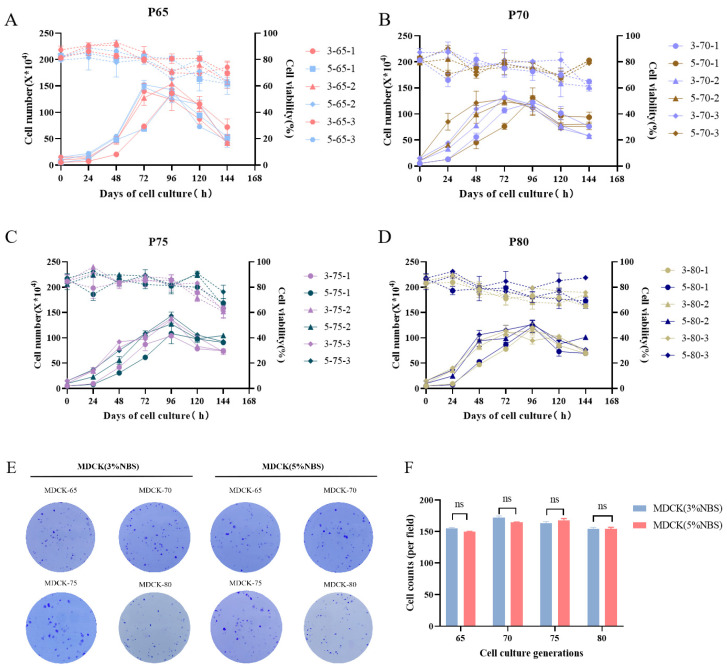
Growth curves of MDCK cells of different generations (**A**–**D**) Solid line: cell density, dashed line: cell viability and cell colony formation (**E**,**F**); numbered as serum content—cell generation—inoculum density, example: 3 (3% NBS)—65 (Cell generation)—1 (Inoculated at a density of 5 × 10^4^ cells/mL). ns Not Significant.

**Figure 5 vaccines-12-00991-f005:**
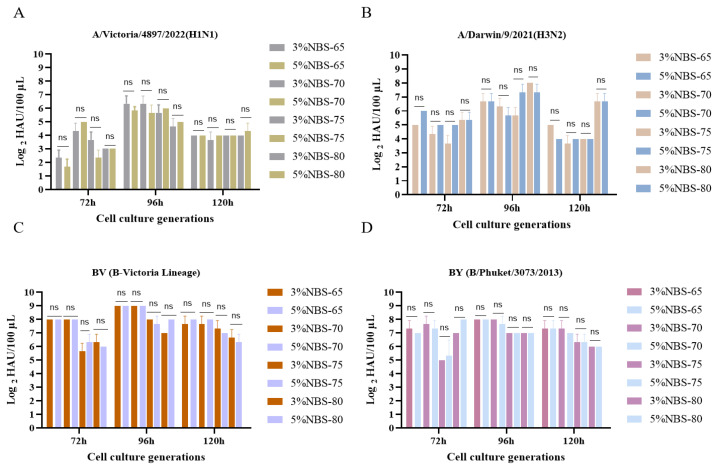
Effect of low-serum medium on the viral susceptibility of different generations of MDCK cells ((**A**) H1N1 virus susceptibility, (**B**) H3N2 virus susceptibility, (**C**) BV virus susceptibility, (**D**): BY virus susceptibility). (3% NBS)—65: (Serum content)-cell generation. ns Not Significant.

**Figure 6 vaccines-12-00991-f006:**
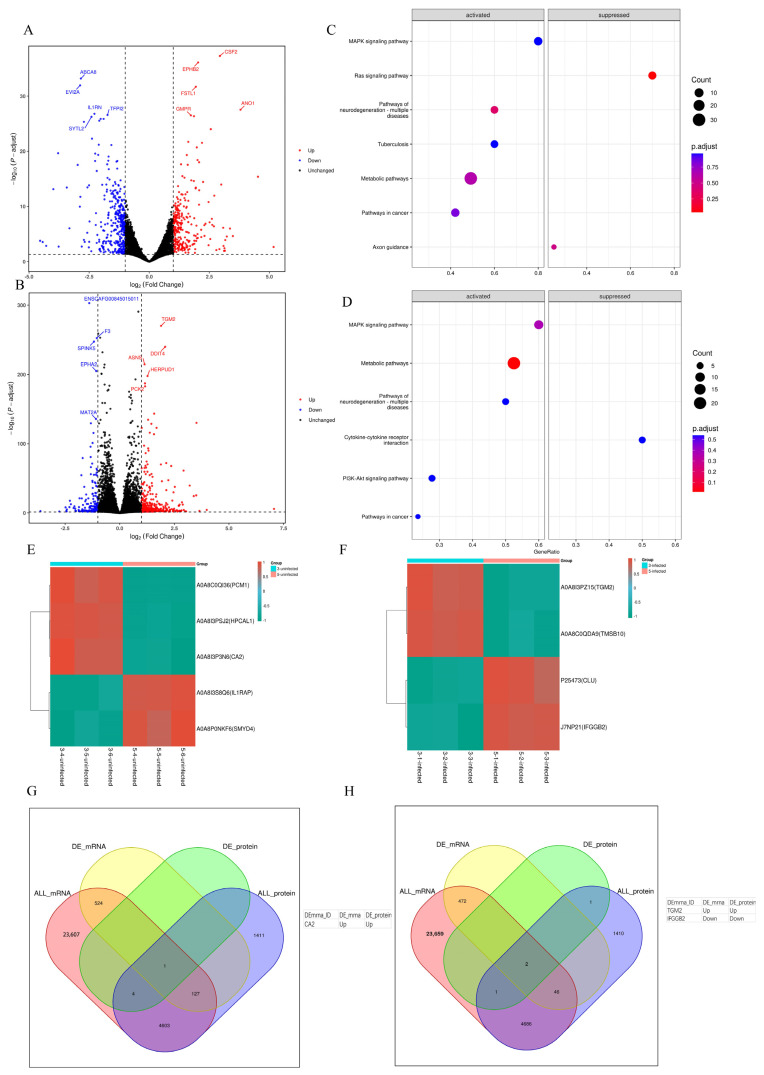
Results of transcriptomics and proteomics analyses. (**A**) Volcano plot showing DEGs of 3-uninfected group vs. 5-uninfected group. Red, blue, and black dots stand for genes with upregulation, downregulation, and non-differentiation, respectively. (**B**) Volcano plot showing DEGs of 3-infected group vs. 5-infected group. (**C**) Dot plot of 3-uninfected group vs. 5-uninfected group KEGG enrichment. (**D**) Dot plot of 3-infected group vs. 5-infected group KEGG enrichment. (**E**) Heatmap showing differentially expressed proteins between the 3-uninfected group and the 5-uninfected group. (**F**) Heatmap showing differentially expressed proteins between the 3-infected group and the 5-infected group. (**G**) Venn diagram showing the results of transcriptomic and proteomic correlation analysis between the 3-uninfected group and the 5-uninfected group. (**H**) Venn diagram showing the results of transcriptomic and proteomic correlation analysis between the 3-infected group and the 5-infected group.

**Table 1 vaccines-12-00991-t001:** Study of cell culture processes in bioreactor with low-serum media.

VP:M199 (NBS)	1:2 (3% NBS)	1:2 (5% NBS)
Inoculation density	80 × 10^4^ cells/mL	80 × 10^4^ cells/mL
Culture time	72 h	72 h
Tank sugar concentration	1.0 g/L	1.0 g/L
Volume of medium	20 L	20 L
Number of cells after digestion	278 × 10^4^/mL × 4.5 L	288 × 10^4^/mL × 4.5 L
	102 × 10^4^/mL × 4.5 L	88 × 10^4^/mL × 4.5 L
	30 × 10^4^/mL × 4.5 L	32 × 10^4^/mL × 4.5 L
Total number of cells digested	410 × 10^4^/mL × 4.5 L	408 × 10^4^/mL × 4.5 L

## Data Availability

All data supporting the findings of this study are available within the manuscript. Any additional data are available from the corresponding author upon reasonable request.

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
