# Peer review of "Production, Passaging Stability, and Histological Analysis of Madin–Darby Canine Kidney Cells Cultured in a Low-Serum Medium"

_vaccines, 2024, doi:10.3390/vaccines12090991_

Round 1

Reviewer 1 Report

Comments and Suggestions for Authors

This manuscript describes the optimization of cell culture medium for MDCK cells. It was found that alternation of neonatal bovine serum (NBS) concentration from 5% to 3% did not significantly affect cell growth and virus productivity.

Considering the application of virus production (described in lines 55-57), it would be more valuable to examine the virus production ability under low serum concentration conditions such as 1% serum and serum-free medium (as a negative control), and compare with those under 3% serum condition.

1) Figure 5: The viral susceptibility should be examined in MDCK cells cultured in a low serum medium (1% or 2%) or serum-free medium. The fact that changing serum content from 5% to 3% did not affect the results is not surprising.

2) Is Chapter 3.4 necessary? It may be necessary to explain why the transcriptomic and proteomic analyses were performed and how they relate to the aims of this study.

Please provide criteria for this judgment that there was no change in protein expression between 3% and 5%.

There is no supplementary data (such as a list of expression levels), so their validity cannot be verified.

3) Lines 158-171: Results should be described based on the data in Fig. 1. There is no explanation for Fig. 1. I could not find data regarding “cell density” (in line 158), “CF1 and CF2 groups” (in line 167), and “overspreading”(in line 169) in Fig.1.

In addition, the figure legend also requires further explanations. It is unclear which data (line and bar charts) corresponds to “Cell number” or “Cell viability”, what the red and blue boxes represent, and what criteria selected those.

Other figure legends have the same problems, which should be revised.

4) Lines 172-174: the sentence can be deleted.

5) Line 139The sentence of “no viral maintenance solution was added” is unclear and may need to explain.

6) Line 66: The reference of “previous study” should be specified.

Author Response

Comments 1: Figure 5: The viral susceptibility should be examined in MDCK cells cultured in a low serum medium (1% or 2%) or serum-free medium. The fact that changing serum content from 5% to 3% did not affect the results is not surprising.

Response 1: I appreciate you asking this question. Based on the relevant literature, using a medium with a specific serum concentration can help the cells attach to the wall as quickly as possible. The medium containing 1%–2% NBS for the MDCK cell culture in this paper has been experimented in the section of 2.1.1, and we found that the cell growth is slow and the state of the cells is poor after amplified culture. Additionally, we have also conducted the relevant experiments and discovered that the cells will not grow normally when the medium does not contain NBS. As a result, the related study will not be conducted later in this paper.(Line 161-Line 168)

Comments 2:

Is Chapter 3.4 necessary? It may be necessary to explain why the transcriptomic and proteomic analyses were performed and how they relate to the aims of this study.

Response : Thank you very much for your careful review. We understand your concerns, as the main function of transcriptomics and proteomics is to compare differences between two samples. However, the results of transcriptomics and proteomics results can offer a more comprehensive insight into the impact of reducing NBS concentration from 5% to 3% on MDCK cell culture. On one hand, the results show that there are only a few differentially expressed genes related to cell growth. The number of differentially expressed proteins does not exceed 10, and further KEGG enrichment analysis did not identify any enrichment of these differentially expressed proteins in any signaling pathway. These results further confirms that alternation of NBS concentration from 5% to 3% did not significantly affect cell growth and virus productivity. On the other hand, the integration of transcriptomics and proteomics enables a comprehensive investigation into other potential differences, such as cell adhesion and tumorigenicity, which are also important for vaccine production. So, the transcriptomic and proteomic analyses were performed.(Line324  -Line327 )

Please provide criteria for this judgment that there was no change in protein expression between 3% and 5%.

Response : Thank you very much for your reminder. Through proteomic analysis, we have indeed discovered the presence of a few differentially expressed proteins, such as PCM1, HPCAL1, CA2, IL1RAP, and SMYD4. Therefore, we believe that the statement 'there was no change in protein expression between 3% and 5%' may not be entirely accurate and delete the related statementsline 276-277. We are so sorry to making this inaccurate description. However, the number of differentially expressed proteins does not exceed 10, and we did not identify any enrichment of these differentially expressed proteins in any signaling pathway through KEGG enrichment analysis. The results suggests that these differentially expressed proteins did not cause differential expression of other proteins in their respective signaling pathways. Therefore, we believe that the impact of individual protein changes on MDCK cells is limited.

There is no supplementary data (such as a list of expression levels), so their validity cannot be verified.

Response : Thank you very much for your suggestion. We have added the relevant data of transcriptome and proteome as Supplementary File 3 in the manuscript. (line 276-277)

Comments 3:Lines 158-171: Results should be described based on the data in Fig. 1. There is no explanation for Fig. 1. I could not find data regarding “cell density” (in line 158), “CF1 and CF2 groups” (in line 167), and “overspreading”(in line 169) in Fig.1.In addition, the figure legend also requires further explanations. It is unclear which data (line and bar charts) corresponds to “Cell number” or “Cell viability”, what the red and blue boxes represent, and what criteria selected those.Other figure legends have the same problems, which should be revised.

Response 3: Thank you very much for your reply, which has been amended accordingly in the original article.Line 173-Line 185

Comments 4:Lines 172-174: the sentence can be deleted.

Response 4:We appreciate your response and have adjusted the location.

Comments 5:Line 139:The sentence of “no viral maintenance solution was added” is unclear and may need to explain.

Response 5:Thank you for your valuable suggestion. The intended meaning of this statement,“no newborn bovine serum is added to the virus maintenance solution” has been revised in the article.(Line 139-Line 142

Comments 6:Line 66: The reference of “previous study” should be specified.

Response 6: I sincerely appreciate your reply. There may be some uncertainties in the article's description, which has been corrected. The content of this article is related to the research of the cell culture process, which is part of the optimization of our prior method. We have not previously published any articles of this module.(Line65-Line 67

Reviewer 2 Report

Comments and Suggestions for Authors

With the work entitled “Production, Passaging Stability, and Histological Analysis of Madin–Darby Canine Kidney Cells Cultured in a Low-Serum Medium” the authors described some experiments that indicate the best conditions to adapt cells to produce influenza vaccines. Through various cellular and molecular tests, they were able to reach certain conclusions that led to positively testing their hypotheses. The professional language in English is consistent, as are the results presented through graphs and images. The hypothesis discussed is in accordance with the data presented and the literature provided at the end of the manuscript.

I have some suggestions and questions.

1. Increase the resolution of figure 1. The graphics are small, as are the color legends.

2. Are there statistical differences between the points present in figures 3, fig 4 (A-D) and fig 5? If yes, authors should add the information.

3. The resolution of figure 6 is poor. The letters are too small.

4. The discussion can be improved. There are few references and a lot of text that just describes what was done in the work, without comparing the data with the vast literature that exists on cell cultivation for vaccine production purposes. All information related to the transcriptome and proteome is missing, which the authors could have used to enrich the discussion.

Author Response

Comments 1: Increase the resolution of figure 1. The graphics are small, as are the color legends.

Response 1: Thank you for pointing this out. I/We agree with this comment. Therefore, we have made some changes to Figure 1 as per advice.(Line 169

Comments 2:Are there statistical differences between the points present in figures 3, fig 4 (A-D) and fig 5? If yes, authors should add the information.

Response 2: First of all, thank you very much for your suggestion; we have performed the relevant statistical analyses in Figure 3 and Figure 5(A-D). Figure 4 (A-D) is the cell growth curve graph, primarily with different densities of cell inoculation, to verify the trend of cell growth in different generations, and to demonstrate that the stability of this cell in low-serum medium is sufficient.Line 210、Line235

Comments 3:The resolution of figure 6 is poor. The letters are too small.

Response 3: Thank you very much for your suggestion. The figure 6 has been modified to optimize the picture font and layout. In case of necessity, higher resolution images will be provided separately. Line 279

Comments 4:The discussion can be improved. There are few references and a lot of text that just describes what was done in the work, without comparing the data with the vast literature that exists on cell cultivation for vaccine production purposes. All information related to the transcriptome and proteome is missing, which the authors could have used to enrich the discussion.

Response 4: Thank you very much for your valuable comments. We have revised the discussion part according to your requirements.(Line 325-366)

Round 2

Reviewer 1 Report

Comments and Suggestions for Authors

The paper has been significantly improved.